# An RNA-Based Vaccine Platform for Use against *Mycobacterium tuberculosis*

**DOI:** 10.3390/vaccines11010130

**Published:** 2023-01-05

**Authors:** Sasha E. Larsen, Jesse H. Erasmus, Valerie A. Reese, Tiffany Pecor, Jacob Archer, Amit Kandahar, Fan-Chi Hsu, Katrina Nicholes, Steven G. Reed, Susan L. Baldwin, Rhea N. Coler

**Affiliations:** 1Center for Global Infectious Disease Research, Seattle Childrens Research Institute, Seattle, WA 98109, USA; 2HDT BioCorp, Seattle, WA 98102, USA; 3Department of Microbiology, University of Washington, Seattle, WA 98109, USA; 4Hematologics Inc., Seattle, WA 98121, USA; 5Department of Pediatrics, University of Washington, School of Medicine, Seattle, WA 98105, USA; 6Department of Global Health, University of Washington, Seattle, WA 98105, USA

**Keywords:** *Mycobacterium tuberculosis*, ID91, GLA-SE, repRNA, nanostructured lipid carrier, tuberculosis, vaccine

## Abstract

*Mycobacterium tuberculosis* (M.tb), a bacterial pathogen that causes tuberculosis disease (TB), exerts an extensive burden on global health. The complex nature of M.tb, coupled with different TB disease stages, has made identifying immune correlates of protection challenging and subsequently slowing vaccine candidate progress. In this work, we leveraged two delivery platforms as prophylactic vaccines to assess immunity and subsequent efficacy against low-dose and ultra-low-dose aerosol challenges with M.tb H37Rv in C57BL/6 mice. Our second-generation TB vaccine candidate ID91 was produced as a fusion protein formulated with a synthetic TLR4 agonist (glucopyranosyl lipid adjuvant in a stable emulsion) or as a novel replicating-RNA (repRNA) formulated in a nanostructured lipid carrier. Protein subunit- and RNA-based vaccines preferentially elicit cellular immune responses to different ID91 epitopes. In a single prophylactic immunization screen, both platforms reduced pulmonary bacterial burden compared to the controls. Excitingly, in prime-boost strategies, the groups that received heterologous RNA-prime, protein-boost or combination immunizations demonstrated the greatest reduction in bacterial burden and a unique humoral and cellular immune response profile. These data are the first to report that repRNA platforms are a viable system for TB vaccines and should be pursued with high-priority M.tb antigens containing CD4+ and CD8+ T-cell epitopes.

## 1. Introduction

For the first time in a decade, 2020 saw an increase in annual deaths (1.5 million total) caused by *Mycobacterium tuberculosis* (M.tb) [1,2,3]. Before the Coronavirus disease 2019 (COVID-19) pandemic, caused by severe acute respiratory syndrome coronavirus 2 (SARS-CoV-2) [4], M.tb was the world’s top cause of death from an infectious disease [5], regaining this title in 2022, with a reported 1.6 million deaths from M.tb [6]. Exposure to M.tb can result in productive infection and pulmonary tuberculosis disease (TB). COVID-19-pandemic-related disruptions in TB care and case finding are estimated by the World Health Organization (WHO) to lead to a further half-million preventable TB deaths [7]. This dual assault from respiratory pathogens is most heavily felt by low- and middle-income countries (LMICs), which endure a disproportionate burden of TB disease [8] and are also suffering from a lack of access to vaccines for SARS-CoV-2 [9]. Moreover, within M.tb, drug resistance is steadily increasing globally. For the past five years, nearly 0.5 million patients infected with M.tb annually develop resistance to front-line antibiotic rifampicin and approximately 80% of those harbor multidrug resistance [1,2,5]. Novel antibiotic regimens [10,11,12,13,14] are being developed but many models are insufficient alone to combat this epidemic [15,16]. Specifically designed low-cost and effective TB vaccines for the prevention of infection (POI) or the complementary prevention of active disease (POD) endpoints are desperately needed [17].

While clinical trials suggest that *Mycobacterium bovis* bacille Calmette-Guérin (BCG) may prevent M.tb infection in adolescents [18], and subunit adjuvanted vaccine candidate M72/AS01_E_ [19,20] prevents TB disease in a subset of interferon-gamma release assay (IGRA)-positive individuals, there is no currently approved vaccine for POI or POD in adults. This lag is partially due to the fact that the financial resources available for testing drug and vaccine candidate regimens in human clinical settings are disproportionately low for TB relative to the global burden of the disease. Addressing the improved efficacy of prophylactic vaccines against M.tb and the supply and costs of manufacturing materials are all required to ensure that a vaccine is readily available to those who need it. Thankfully, renewed enthusiasm for TB vaccine research has been stimulated largely by two main pillars of preclinical and clinical advancements. First, the pursuit of correlates of protection against TB, which are refining the immune players in the fight against infection and disease, and second, the pursuit of cutting-edge vaccine delivery platforms.

Historically the vaccine pipeline has relied on CD4+ T helper 1 (TH1) type responses as a benchmark for the immunogenicity stage-gating of vaccine candidates. However, the full mechanism of protection has yet to be determined [21], and several preclinical and clinical reviews suggest that the CD4+ TH1 subset producing proinflammatory cytokine IFNγ alone is not sufficient nor fully predictive of clinical efficacy [22,23,24,25]. Indeed, protection from M.tb infection and TB disease is likely a multifaceted process involving many cell types beyond canonical CD4+ T cells, and our focus here is on M.tb-antigen-specific CD8+ T cells. Many reviews are devoted to the significance of CD8+ T cells in TB infection and disease [26,27,28,29,30,31], yet still, they are not prioritized as a vaccine target cell type in most candidate screens. Importantly, CD8+ T cells can migrate to the site of M.tb infection [32,33,34], and removing MHC class I or CD8+ T cells in vivo enhances M.tb susceptibility [29,35,36,37,38]. Cytolytic CD8+ T cells produce IFNγ, a key proinflammatory cytokine known to help control M.tb and lyse M.tb-infected macrophages [39,40,41,42]. Due to their ability to home to pulmonary spaces and anti-M.tb-effector functions, it is unsurprising that in preclinical models of an M.tb challenge, CD8+ T cells help reduce bacterial burden [26,29,33,43]. Most recently, the use of intravenous (i.v.) BCG vaccination in nonhuman primates (NHP) has provided a benchmark of immune responses that correlated with nearly sterilizing protection from an M.tb challenge [44]. Notably, in the routine intradermal (i.d.) BCG vaccine NHP cohort, there was a lack of proinflammatory CD8+ T cells and less protection when compared to i.v. BCG [44]. These data collectively suggest that CD8+ T cells represent an underappreciated target for vaccine-induced efficacy endpoints.

Many strategies designed to induce robust anti-M.tb CD8+ T cells have been developed and are well-reviewed here [26,27,31]. Some examples include recombinant BCG (VPM1002), chimpanzee adenoviral vectored strategies (ChAdOx185A), modified vaccinia virus Ankara (MVA85A), recombinant adenoviruses (Ad5Ag85A, AERAS-402), and Cytomegalovirus vector approaches (CMV-6Ag and MTBVAC) and are under clinical evaluation [21,26,27,45,46]. Interestingly, a majority of these candidates are vector- or replication-based. Enriching the pipeline with novel candidates and platforms is warranted, given the expanding immune correlates being discovered and the need for broad accessibility in LMICs. Leveraging RNA vaccines, for example, circumvents the challenges of protein manufacturing, reducing vaccine development timelines from several years to months and reducing costs throughout pipeline evaluations and deployment. For example, once the SARS-CoV-2 sequence was made available publicly, messenger RNA (mRNA) vaccines were developed within months and tested in human clinical trials [47]. While RNA vaccine platforms have been very minimally leveraged for TB vaccines [48] in preclinical efficacy testing to our knowledge, the evidence for targeting multifaceted immune responses against M.tb for high vaccine efficacy provides a good rationale for evaluating this platform. Indeed BioNTech (Mainz, Germany) has recently announced the initiation of a phase-one clinical trial for the safety testing of an mRNA TB vaccine candidate in collaboration with the Bill and Melinda Gates Foundation (ClinicalTrials.gov Identifier: NCT05547464). We and others have previously demonstrated robust humoral and cellular responses, including CD8+ T-cell responses, induced by replicating RNA delivered via multiple different modalities [49,50,51,52]. For these reasons, we designed and evaluated a novel replicating RNA platform (repRNA) [49] both as a homologous TB vaccine and as a heterologous strategy to complement protein–adjuvant prophylactic vaccine candidates against an M.tb challenge in mice. We hypothesize that the replicating aspect of repRNA, resulting in longer antigen exposure [49] more similar to BCG than protein-adjuvant strategies, will enhance immune responses against selected M.tb vaccine candidate antigens.

## 2. Materials and Methods

*ID91 repRNA production and qualification*. The codon-optimized sequence encoding ID91 was synthesized (Codex DNA) and cloned into the Venezuelan equine encephalitis virus replicon (strain TC83) between PflFI and SacII sites using a Gibson assembly. The sequence-verified DNA was then linearized by NotI digestion and RNA transcribed and capped by T7 RNA polymerase and Vaccinia capping enzyme reactions, respectively, as previously described [50]. To validate the ID91 repRNA produced ID91 protein, Baby Hamster Kidney (BHK) cells (ATCC) were transfected with 4 µg ID91-repRNA or GFP-repRNA as a negative control using lipofectamine 2000 (ThermoFisher, Waltham, MA, USA). After 24 h, the cells and supernatants were collected for analysis by SDS-PAGE and Western blot under reducing conditions. Included in the blot were ID91 protein control and MagicMark XP ladder (Thermo Fisher). For ID91 antigen detection after the transfer to the PVDF membrane, primary mouse sera (from animals immunized 3 times 3 weeks apart with ID91 + GLA-SE) was used at 1:500 in PBS. Goat anti-mouse HRP (SouthernBiotech) 1:10000 was used for detection along with SuperSignal™ West Pico PLUS Chemiluminescent Substrate (ThermoFisher). The resulting image was captured on a Biorad chemidoc XRS+.

*Preclinical animal model.* Female C57BL/6 mice 4–6 weeks of age were purchased from Charles River Laboratory. The mice were housed at the Infectious Disease Research Institute (IDRI) biosafety level 3 animal facility under pathogen-free conditions and were handled in accordance with the specific guidelines of the IDRIs Institutional Animal Care and Use Committee. The institute, formerly known as IDRI, operated under USDA Certificate # 91-R-0061 and PHS Assurance # A4337-01. The mice were infected either with a low-dose (50–100 bacteria) aerosol (LDA) or ultra-low dose (1–8 bacteria) aerosol (ULDA) of M.tb H37Rv using a University of Wisconsin-Madison aerosol chamber. Twenty-four hours post-challenge, the lungs of 3 mice were homogenized and plated on Middlebrook 7H11 agar (Fisher Scientific) to confirm delivery.

*Vaccines and adjuvants*. The cohorts of mice were immunized intramuscularly (i.m.) twice three weeks apart for epitope mapping, once 6 weeks before the challenge, or twice three weeks apart, with the final immunization occurring 3 weeks before the challenge for immunogenicity and efficacy testing. The mice received either saline alone or vaccinations containing 0.5 µg/dose of ID91 recombinant fusion protein combined with 5.0 µg/dose GLA-SE, as previously published [53,54,55]. A separate cohort of mice was immunized with 1.0 µg repRNA complexed with NLC, as described [49]. Homologous or heterologous regimens were also leveraged using the doses and regimens outlined above. Combination cohorts received ID91 + GLA-SE in the right hind limb followed by ID91 repRNA + NLC in the left hind limb at both prime and boost time points in the same doses described above.

*Epitope mapping:* Two weeks post-boost, the splenocytes were isolated from the cohorts of ID91 + GLA-SE or ID91 repRNA + NLC-immunized mice and stimulated with ID91 protein antigen (10 µg/mL), overlapping peptides (10 µg/mL), or media alone for 60 h at 37 °C and 5% CO_2_ with proliferation dye (Violet Proliferation Dye, BD Biosciences). The peptides were generated in the 15 mer format with 8 amino acid overlaps for the ID91 fusion antigen, for a total of 125 peptides (BioSynthesis). These 125 peptides make up 13 different pools of 5–10 peptides each (Table 1). Cells were then restimulated for two hours, brefeldin A was added directly to the plate and cells were allowed to incubate for a further 4 h. The cells were then washed, and ICS and flow cytometry analysis were performed, as described below.

*Flow cytometry.* Intracellular flow cytometry was performed on the splenocytes post-immunization, but the pre-challenge and lung homogenates post-infection. The samples were incubated, washed, and stimulated, as previously described [54]. The cells were stimulated with media alone, 10 µg/mL of recombinant ID91, or 1 µg/mL phorbol myristate acetate (PMA) (Calbiochem) +1 µg/mL ionomycin (Sigma-Aldrich). Some sample stimulations also contained fluorescently labeled anti-mouse CD107a (1D4B, BioLegend). After stimulation, the samples were stained for markers of interest using fluorescent-conjugated antibodies, as previously described [54]. Notably, all of the sample stains were used at 10 µL/mL concentration. The primary surface staining included: anti-mouse CD4 (clone RM4-5, BioLegend), CD8a (clone 53–6.7, BioLegend), CD44 (clone IM7, eBioscience), CD154 (clone MR1, BioLegend), and 1 µg/mL of Fc receptor block anti-CD16/CD32 (clone 93, eBioscience) in PBS with 1% bovine serum albumin (BSA). The cells were then washed and fixed using BD Biosciences Fix/Perm reagent for 20 min at RT. Intracellular staining was carried out in Perm/Wash (BD Biosciences) reagent with anti-mouse GMCSF (clone MP1-22E9, BioLegend), IFN-γ (clone XMG1.2, Invitrogen), IL-2 (clone JES6-5H4, eBioscience), IL-17A (clone Tc11-1BH10.1, BioLegend), TNF-α (clone MP6-XT22, eBioscience), and IL-21 (mhalx21, eBiosciences) for 10 min at RT. The samples were then incubated in 4% paraformaldehyde for 20 min to fix, decontaminate, and remove from the containment facility before washing and resuspension in PBS + 1% BSA. An LSRFortessa flow cytometer (BD Bioscience) was used for sample acquisition, and analysis was performed using FlowJo v10.

*Endpoint titer (EPT) ELISA.* The serum was collected from terminal bleeds immediately after euthanasia at the post-boost or post-infection time points. Bronchoalveolar lavage fluid (BALf) was collected at the post-infection time points immediately after euthanasia. The serum samples were diluted to 1:100, and the BALf samples were added directly (neat) to plates coated with 2 µg/mL of ID91, Rv1886, Rv2389, Rv3478, or Rv3619. After overnight incubation, the plates were exposed to HRP-conjugated anti-IgA, Total IgG, IgG1, or IgG2c (Southern Biotech). The plates were then developed with a tetramethylbenzidine substrate and stopped with 1 N H_2_SO_4_. The plates were read at 450 nm with 570 nm background subtraction. The EPTs were calculated using a regression analysis of the sample dilution and O.D. on GraphPad Prism.

*Bacterial burden/Colony-forming units (CFU).* Then, 24 h to 3 weeks post-infection with M.tb, 3–7 mice per group were euthanized using CO_2_. The lung and spleen tissue from the infected animals was isolated and homogenized in 5 mL of either RPMI + FBS (lung) or PBS+ Tween-80 (Sigma-Aldrich, St. Louis, MO, USA) CFU buffer (spleen) using an Omni tissue homogenizer (Omni International, Kennesaw, GA, USA). Serial dilutions of homogenate were made in CFU buffer, and the aliquots were plated on Middlebrook 7H11 agar plates and subsequently incubated at 37 °C and 5% CO_2_ for 2–3 weeks before the colonies were counted. Bacterial burden, as CFU/mL, was calculated per organ and is presented here as Log10 values. Reduced bacterial burden was calculated as the difference in the mean Log10 values between the groups assessed.

*Statistics.* The number of animals included in the efficacy studies (*n* = 7 mice/group) is required to provide a statistical power of 95% with an alpha (*p*-value) of 0.05 to detect a significant difference in the bacterial burden between the groups by the CFU-plating of the lung homogenates post-challenge. The bacterial burden and humoral and cellular immune responses were assessed using a One-way ANOVA with Dunnets multiple comparison test between the vaccinated groups and the saline control. The flow cytometry data were assessed using FlowJo v10 (BD), and SPICE (NIH) using the Wilcoxon signed rank test as compared to the untreated groups or One-way ANOVA with Dunnets multiple comparison test. Statistical analyses were performed using GraphPad Prism 7 software. Significant differences are labeled accordingly in the figures where * *p* < 0.05, ** *p* < 0.01, *** *p* < 0.001, and **** *p* < 0.0001, with the methodology used previously by our group [54,56].

## 3. Results

### 3.1. M.tb RNA Candidate Vaccine

The preclinical M.tb vaccine candidate ID91 is a fusion of four M.tb proteins: Rv3619 (esxV; ESAT6-like protein), Rv2389 (RpfD), Rv3478 (PPE60), and Rv1886 (Ag85B) (Figure 1A). These antigens were selected as candidates for M.tb vaccines based on their lack of human-sequence homology and their ability to induce an ex vivo IFN-γ response in PPD+ human peripheral blood mononuclear cells (PBMC), suggesting that they are immunogenic in humans [57,58]. ID91 antigens were cloned into an alphavirus repRNA backbone, as previously described [49] (Figure 1B). This repRNA platform has a number of advantages, including (1) a subgenomic promoter to amplify the expression of the antigen of interest, (2) exclusive cytoplasmic activity with no risks of nuclear genomic integration, and (3) sustained antigen production in the presence of adjuvating innate responses that allow for dose-sparing and strong cellular responses [49,50]. Using Western blotting, we observed the efficient expression of the ID91 antigens from repRNA ID91 in BHK cells (Figure 1C and Appendix A), demonstrating that this platform is amenable to M.tb antigen expression. Partnering repRNA with delivery formulations provides stability and protection from RNAses and removes the requirement for viral delivery, eliminating anti-vector immunity, which has been detrimental for promising candidates.

Here, we leveraged a first-generation nanostructured lipid carrier (NLC) [49] as a two-vial approach with ID91-repRNA. The C57BL/6 mice were vaccinated two times three weeks apart with either our existing next-generation ID91 fusion protein [57] +TLR4 agonist, glucopyranosyl lipid adjuvant formulated in a stable emulsion (GLA-SE) as a positive control for CD4+ T-cell responses and prophylactic efficacy against M.tb or a novel candidate ID91-repRNA + NLC to evaluate rep-RNA induced preliminary immunogenicity (Figure 2A). Splenocytes from both immunized groups were stimulated two weeks post-boost with media alone, ID91 protein or each of 13 different 15 mer peptide pools overlapping by 8 mers of ID91 and evaluated for proliferation and cytokine responses (Figure 2B, representative gating in Appendix A). We observed that ID91 + GLA-SE elicited both CD4+ (Red) and CD8+ (Blue) T-cell responses, albeit larger CD4+ TH1, while ID91-repRNA + NLC largely drove CD8+ T-cell proliferation and inflammatory cytokines (Figure 2B). For the ID91 protein/GLA-SE vaccine, we observed both CD4+ and CD8+ epitopes in the ID91 antigen fusion. Vaccination with ID91 + GLA-SE elicited measurably higher proliferative CD4+ T-cell responses from peptide pool numbers 2, 4, 10, 12, and 13, whereas pools 3, 4, 10, 11, and 12 induced measurable proliferation in the CD8+ T cell compartment (Figure 2B). Amino acid sequence LVAAAKMWDSVASDLFSAASAFQSVVWGL (peptides #33–35, pool 4) from Rv3478 may contain dominant epitopes for both CD4+ and CD8+ T cells. Unlike the protein-adjuvant strategy, repRNA ID91 + NLC induces measurable CD8+ but not CD4+ T-cell proliferation and cytokine induction upon restimulation of splenocytes ex vivo, with the most dominant epitopes in Rv1886 (Figure 2B, Table 1).

### 3.2. Immunogenicity and Prophylactic Protective Efficacy

We first evaluated ID91 repRNA and ID91-GLA-SE prophylactic immunizations in C57BL/6 mice for immunogenicity as well as protection from infection six weeks after a single immunization as a stringent criterion for advancing this platform (Figure 3A). A single vaccination with ID91 + GLA-SE induced robust CD4+ CD44+ TH1 T cells expressing IFNγ, IL-2, and TNF (Figure 3B), but negligible CD8+ CD44+ cytokine-producing T cells (Figure 3C and Appendix A). Conversely, ID91 repRNA + NLC trended (but not significant) towards the higher induction of IFNγ, IL-2 or TNF-producing CD8+ CD44+ T cells (Figure 3C) as well as moderate generation of TH1 CD4+ T-cell responses (Figure 3B). No significant IL17a or IL-21 expression was observed for any group. Both of the vaccines induced a robust total IgG humoral immune response to the ID91 fusion, significantly greater than the saline group but not significantly different between the two platforms, with the bulk of this response being against antigen Rv1886 (Figure 3D). Lastly, we observed that both ID91 + GLA-SE and ID91 repRNA + NLC reduced lung bacterial burden, 0.55 and 0.43 log protection versus saline, respectively, three weeks post-challenge with a low-dose aerosol (LDA 50-100 colony-forming units [CFU]) of M.tb H37Rv (Figure 3E). These data demonstrate that the repRNA platform is immunogenic and affords some prophylactic protection in a stringent preclinical mouse efficacy screening model of TB.

Next, we evaluated the protein- and RNA-based platforms for immunogenicity and protection against an ultra-low dose (ULDA, 1–8 bacteria) of the H37Rv challenge using heterologous, homologous, and combination prime-boost strategies (Figure 4A). For the combination strategies, the mice were given both ID91 + GLA-SE and ID91 repRNA + NLC i.m. in separate limbs at prime and boost time points to avoid potential innate immune-mediated interference between the two modalities. We hypothesized that driving CD4+ and CD8+ T-cell responses may have an additive or synergistic effect on immunogenicity and provide subsequent protection from the more physiologically relevant ULDA challenge. We observed that two immunizations with ID91 + GLA-SE enhanced protection in the lung (0.743 log10) over a single dose (Table 2, Figure 4B), which is in line with prior publications from our group [59]. However, two immunizations with ID91 repRNA + NLC only afforded a moderate reduction in bacterial burden in the lung, 0.306 log10 (Table 2, Figure 4B), which we believe may be due to the suboptimal timing between the prime and boost vaccinations based on recent observations outside the scope of this manuscript. Heterologous strategies also differed, with RNA-prime protein-boost being the most efficacious regimen evaluated for reduced lung bacterial load (0.847 log10 reduction), followed closely by the combination regimen (0.809 log10 reduction) (Table 2, Figure 4B). While the combination regimen appears to be the most protective from dissemination to the spleen (1.419 log10 reduction from saline), this was not statistically significant due to a wide standard error (Table 2, Figure 4C).

This pattern of efficacy is largely mirrored in the post-challenge humoral and cellular immunogenicity readouts. While i.m. vaccine strategies are not known for developing robust mucosal immune responses, we evaluated bronchoalveolar lavage fluid for anti-ID91 IgA responses at 3 weeks post-challenge. Not surprisingly, the magnitude of IgA produced by all regimens was low, and homologous ID91 + GLA-SE was the only strategy to induce a statistically significant response compared to saline (Figure 4D). We observed that the combination and homologous ID91 + GLA-SE regimens induced the highest ID91-specific total IgG endpoint titers in serum (EPT) post-challenge, 7.45 log10 and 8.25 log10, respectively (Table 2, Figure 4E). They are followed by heterologous regimens, while the lowest total IgG induced was in the homologous RNA cohort (Table 2, Figure 4E). Similar trends were observed for ID91-specific IgG1 (Figure 4F) and IgG2c (Figure 4G), with homologous protein and combination regimens demonstrating the highest EPT and homologous RNA the lowest EPT.

Interestingly we did not observe significant differences in the ID91-specific stimulation of the individual cytokines IL-2, GM-CSF, or IL-17A from CD4+ nor CD8+ T cells between the treatment groups. However, CD4+ T cell TNF expression was statistically higher in the heterologous RNA prime, protein boost, and combination regimens, and CD8+ T cell TNF expression was significantly higher in the combination regimen. Given the relative paucity of strong CD8+ epitopes in ID91 antigens, it was not a surprise to observe the expression of CD107a on CD8+ CD44+ T cells post-challenge to be statistically similar between the cohorts. We also examined polyfunctional CD4+ and CD8+ T-cell responses by using flow cytometry results from each cohort after the ex vivo stimulation of the lung cells and found that only the combination regimen induced a significantly higher total magnitude of responses for both the CD4+ and CD8+ subsets (Figure 4H–K, Table 2, representative gating Appendix A). We observed the greatest magnitude and diversity of responses for CD4+ T cells in the context of protein stimulation and for CD8+ T cells in the context of peptide pool stimulation. While not significant, there is a trend for the most robust polyfunctional CD4+ T-cell responses in regimens that include at least one arm, including protein immunizations. We observed the composition of polyfunctional cells to be relatively similar (Figure 4H–K). Triple-positive (IFNγ, IL-2, and TNF), dual-expressing IFNγ+ TNF+, and single-TNF-positive cells made up the majority of the CD4+ T-cell responses across the regimens evaluated, with either protein or peptide stimulations. While single-positive IL-2+ or TNF+ responses overwhelmingly dominated the CD8+ T-cell response in this post-challenge ex vivo stimulation with either protein or peptide pools (Figure 4H–K). The homologous RNA group and combination groups seem to have the most diverse CD8+ T-cell responses when restimulated with peptide pools, but these are largely lost in the context of full-protein stimulation (Figure 4J,K).

## 4. Discussion

These data provide the first report of a replicating RNA-based vaccine strategy being evaluated in the M.tb mouse challenge model and showing efficacy in heterologous and combination regimens. While ID91 served here as a proof-of-concept antigen that afforded some protection from the bacterial burden, epitope mapping demonstrated that ID91 contains few dominant CD8+ T-cell epitopes in the C57BL/6 mouse model and is, therefore, not a high-priority candidate for the repRNA platform. Future work that incorporates antigens containing CD8+ T-cell epitopes [30,60,61] should be prioritized and include specific mechanistic studies that evaluate the contribution of CD8+ T cells elicited by the RNA platform. The ability to readily swap antigens lends itself well to the needs of TB vaccines since global regions harbor different predominant lineages [62], and tailoring vaccines to meet regional needs may further help improve efficacy. Indeed, while we observed modest protection from bacterial burden in the lungs of homologous protein and heterologous candidate vaccinated mice, there was an absence of protection observed in the spleen at this timepoint evaluated. Improving protection from dissemination should be an additional selection criterion for novel antigen selection.

While there is some overlap in the CD4+ and CD8+ responses driven by each immunization strategy, these data demonstrate that RNA and protein differentially elicit CD4+ and CD8+ T-cellular responses to distinct antigen epitopes (Figure 2B), which may be beneficial for heterologous vaccine approaches. This is congruent with clinical trials of a similar but more advanced clinical vaccine candidate, ID93 + GLA-SE, where we have observed protein-adjuvant vaccine-elicited immunogenicity to be dominated by CD4+ TH1 responses and far fewer cases (four responders of 38 across all of the regimens tested, ClinicalTrials.gov number NCT02465216) demonstrating CD8+ T-cell-based responses [63].

We observed that the heterologous and combination regimens were moderately additive in reducing pulmonary bacterial burden in this preclinical model. Importantly we observed that the repRNA platform induced immunogenic responses to ID91 antigens both post-vaccine boost as well as the post-M.tb challenge. These were measured by robust systemic antigen-specific antibody responses and some evidence of mucosal-driven immunity with the detection of ID91-specific IgA in the BALf of the immunized animals compared to the saline mouse cohorts. We also note that the cellular CD4+ and CD8+ T-cell responses were differentially induced by each platform, and the polyfunctional responses were disparate depending on the regimen being evaluated in the post-challenge time point. Those regimens containing a repRNA arm did display enhanced magnitude or composition of polyfunctional responses post-challenge than saline or protein-only controls. These data suggest that the repRNA vaccine is eliciting a measurably broader CD8+ T-cell response, even in the condition of an antigen with few epitopes.

It is interesting that the heterologous protein > repRNA cohorts and repRNA > protein cohorts did not behave similarly in the protection or immunogenicity readouts. In fact, we observed that the repRNA > protein regimens were more protective (lung bacterial burden) and induced more polyfunctional CD4+ T cells than protein > RNA regimens, albeit these differences were small. We speculate this may have to do with the repRNA > protein regimen allowing for the establishment of CD8+ T-cell responses in tandem with CD4+ responses instead of a dominating CD4+ response that takes over at the expense of resources for the CD8+ T cells. More work to define these properties and establish this mechanism is needed. While no single immunological endpoint (total IgG, IgG1, IgG2c, IgA EPT, TH1 CD4+ or CD8+ T cells) correlated with protective efficacy, there are interesting trends that may be further explored and optimized. Indeed, our data suggest that meeting the thresholds for specific combinations of responses versus single endpoints may better correlate with protective efficacy in this model.

## 5. Conclusions

The landscape of TB vaccines is ready for a dramatic surge forward. A key advantage of RNA platforms is their ability to stimulate antigen-specific CD8+ T cells and antibody responses, which both aid in the control of intracellular infections. As a result, heterologous and co-immunization strategies that combine protein and nucleic acid vaccines are promising approaches for inducing protective immunity through CD4+ and CD8+ T cell-mediated responses [64,65]. In our observations, we also saw that polyfunctional CD4+ T-cell responses were generated in the heterologous regimens, so the inclusion of repRNA will not be to the detriment of this important cell type for TB control.

Future interrogations should examine other immune correlates that are showing promise, including IgM, as well as the localization and kinetics of CD4+ and CD8+ T-cell responses [44,66]. The mucosal or aerosol delivery of repRNA in specialized delivery formulations may also help drive pulmonary-specific immunity against M.tb, and these strategies are currently being evaluated. Research outside the scope of this work is currently ongoing to formulate repRNA for aerosol or i.n. delivery. We hypothesize that leveraging repRNA in a heterologous prime-pull technique may provide the balance of durable systemic and mucosal immunity, which is showing promise in recent preclinical models as correlates of protection [44]. Our prior work with protein+ adjuvant candidates has shown that intranasal administration retains protective efficacy in the mouse model and enhances the induction of tissue-resident TH17 cells [67], which also may contribute to TB control. The optimal heterologous regimen composition, route, and regimen are outside the scope of this study, but they are ongoing questions in our laboratories.

Evaluating alternative repRNA expression strategies and alternative timing between prime and boost immunizations can be explored to optimize B- and T-cell responses against antigens of interest. For example, here we evaluated a prime/boost spaced 3 weeks apart in our ULDA challenge and have more recently established that 4–8 weeks between prime and boost drives better immune responses in regimens that include the repRNA platform (unpublished and [50]). We believe that the moderate additive effect seen here with the heterologous and combination regimens, compared to the LDA study with a 6-week rest before the challenge, could be significantly improved by increasing the vaccine interval timing. Here, we examined LDA and ULDA challenges with laboratory-adapted M.tb H37Rv; follow-on work should evaluate the efficacy against clinical isolates representing major M.tb lineages.

While this preclinical proof of concept work leveraged an NLC formulation, next-generation RNA vaccine formulations have advanced into clinical trials (e.g., HDT BioCorp: state-of-the-art delivery vehicle Lipid InOrganic Nanoparticle (LION) [50], US ClinicalTrials.gov Identifier: NCT04844268, India CTRI/2021/04/032688) and the development of RNA-based M.tb vaccines should engage these formulations or others [68,69,70,71,72] with proven human safety data to help accelerate this platform and address the ongoing TB epidemic. In summary, the repRNA platform shows promise as an immunogenic vaccine strategy for TB, and a well-designed set of M.tb antigens in partnership with advanced RNA formulations have the capability to significantly contribute to the TB vaccine pipeline.

## Figures and Tables

**Figure 1 vaccines-11-00130-f001:**
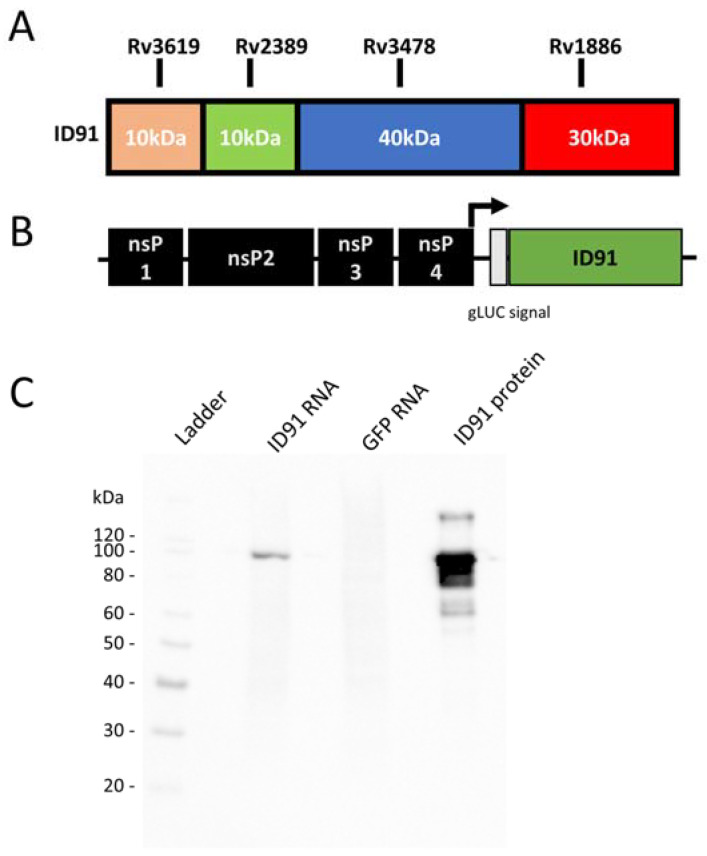
Design and Characterization of ID91 repRNA vaccine candidate. (**A**) ID91 fusion protein comprising 4 M.tb Antigens Rv3619, Rv2389, Rv3478, Rv1886. (**B**) Alphavirus backbone expressing nonstructural proteins (nsP) and ID91 fusion antigen. (**C**) Western blot of ID91 antigen expression from transfected BHK cell lysate in vitro after 24 h. From left to right: Ladder, Cell lysate from BHK transfected with ID91 RNA, GFP RNA, ID91 protein.

**Figure 2 vaccines-11-00130-f002:**
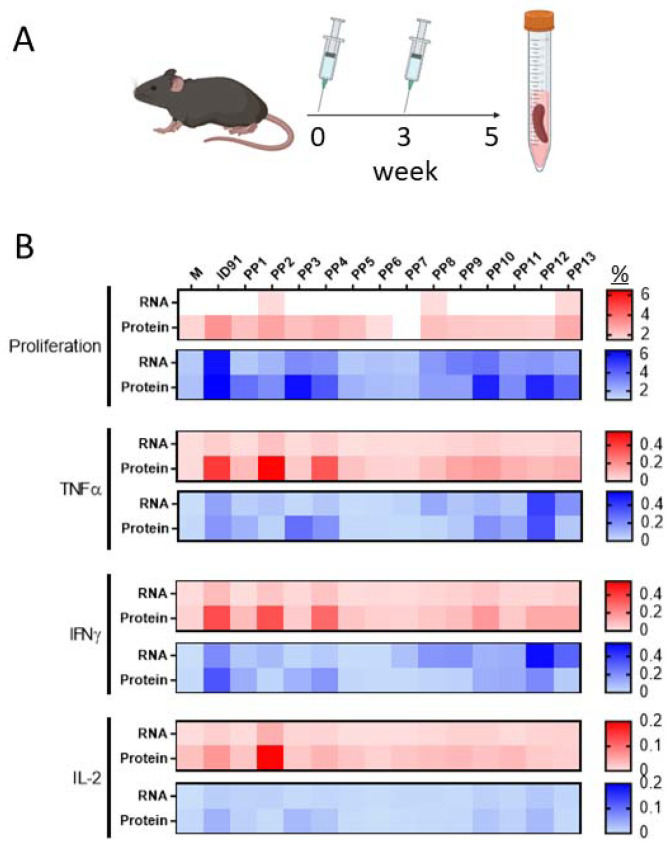
Epitope mapping of T-cell responses with different vaccine platforms. (**A**) Vaccination scheme. C57BL/6 mice were immunized twice, three weeks apart, with ID91 repRNA + NLC (RNA) or ID91 + GLA-SE. (**B**) Two weeks post-boost, splenocytes were stimulated with media (M) whole antigen (ID91) or 13 different peptide pools (PP) with overlapping 15 mers of ID91. Proliferation and cytokine expression were measured by flow cytometry after stimulation. Heat map depicts the percentage of CD4+ (red)- or CD8+ (blue)-responding T cells from each immunization depicted to the left (RNA or Protein). Data representative of a single experiment with *n* = 4 animals per group.

**Figure 3 vaccines-11-00130-f003:**
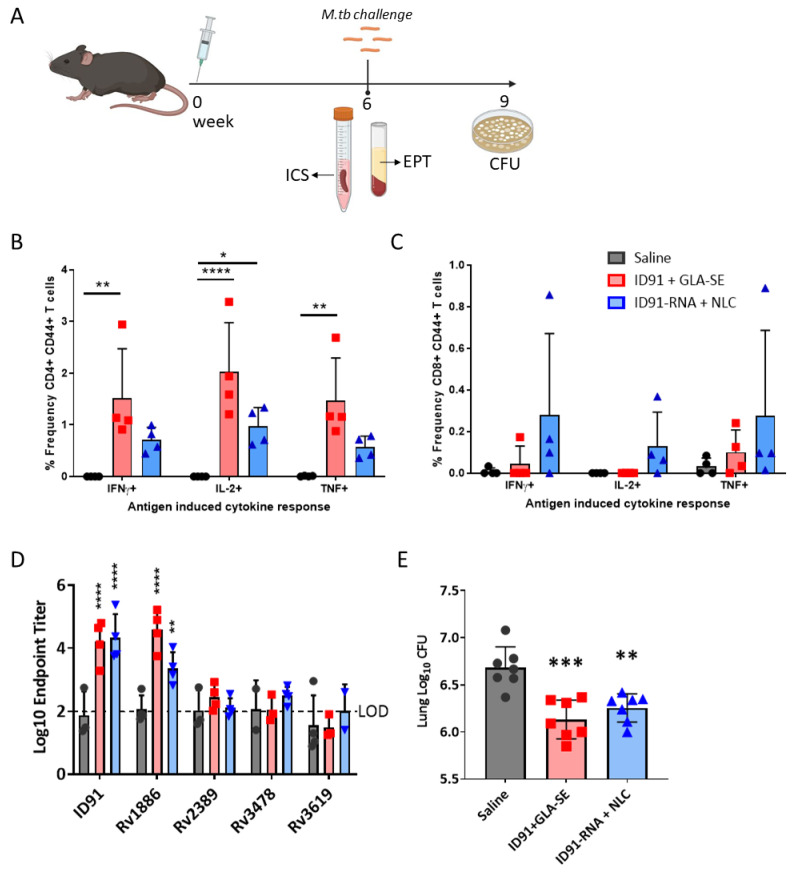
Single immunization with repRNA vaccine is moderately immunogenic and affords prophylactic protection. (**A**) Experimental schematic showing splenocytes from cohorts of saline (grey circles), ID91 + GLA-SE (red squares) or ID91 repRNA + NLC (blue triangles) immunized mice 6 weeks post-vaccination were stimulated with ID91 ex vivo and evaluated by intracellular cytokine staining flow cytometry including (**B**) percentage of CD4 + CD44 cytokine-producing T cells as well as (**C**) percentage of CD8+ CD44+ cytokine-producing T cells after medium subtraction. *n* = 4 per group. (**D**) Plasma was evaluated 6 weeks post-vaccination for total IgG responses to fusion antigen ID91 and its components, Log10 endpoint titer (EPT) is shown. *n* = 4 per group. (**E**) Pulmonary bacterial burden Log10 CFU 3 weeks post-LDA challenge with M.tb H37Rv. *n* = 7 per group, mean value and SEM shown with Y-axis beginning at 5.5 Log10 CFU. Asterisks represent a statistically significant difference from saline using One-way ANOVA with Dunnett’s multiple-comparison test. All data are representative of two independent repeated experiments. * *p* < 0.05, ** *p* < 0.01, *** *p* < 0.001 and **** *p* < 0.0001.

**Figure 4 vaccines-11-00130-f004:**
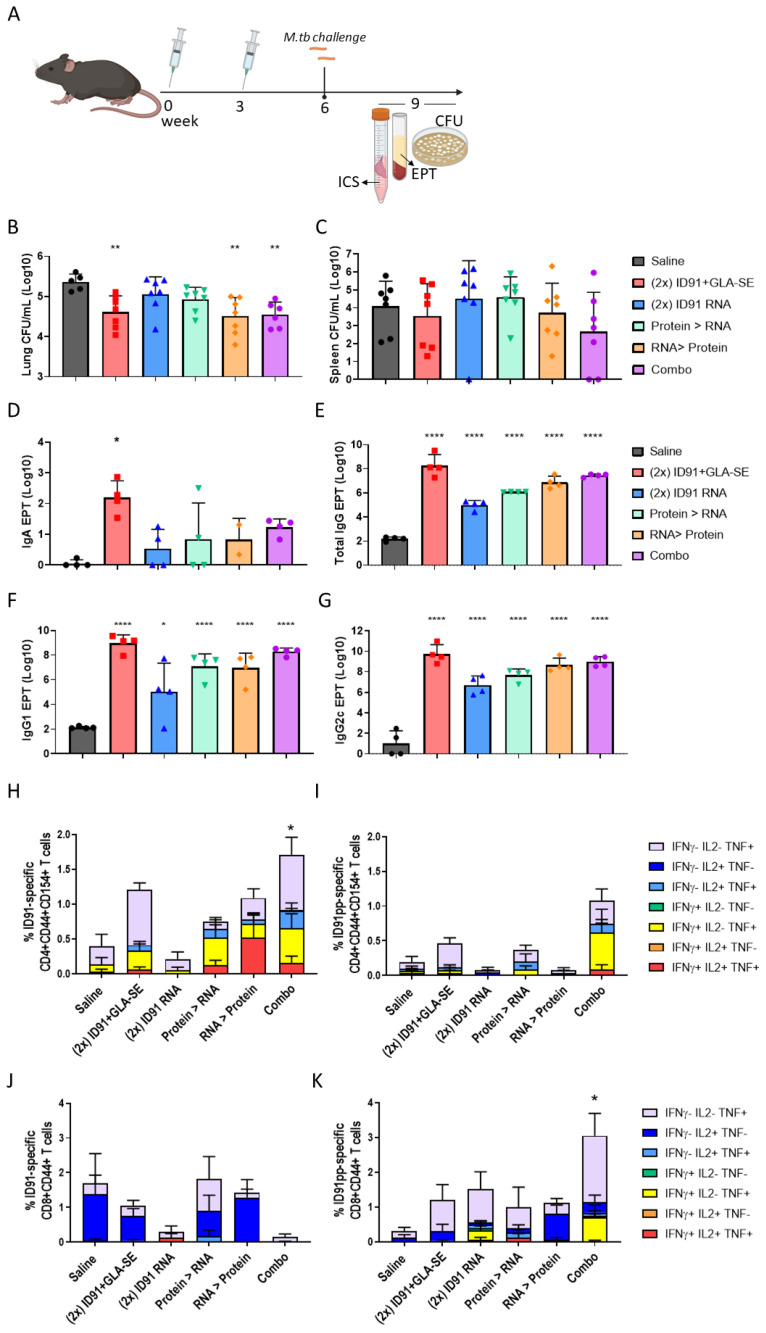
Regimen composition influences protection and post-challenge immune responses. (**A**) vaccination and sample schematic. Bacterial burden assessed by colony-forming units (CFU) in (**B**) the lung and (**C**) spleen homogenates 3 weeks post-challenge. *n* = 6–7 mice per cohort, saline–grey, homologous ID91 + GLA-SE–red, homologous ID91 repRNA +NLC–blue, protein prime/RNA boost–green, RNA prime/protein boost–orange, and combination–purple. Y-axis for (**B**) does not begin at zero. (**D**) BALf from *n* = 4 mice per cohort was collected 3 weeks post-challenge and evaluated for ID91-specific IgA responses. Plasma from *n* = 4 mice per cohort was isolated 3 weeks post-challenge and evaluated for ID91-specific (**E**) Total IgG, (**F**) IgG1 and (**G**) IgG2c. CFU and EPT group means were compared to saline by ordinary One-Way ANOVA and Dunnet’s multiple comparisons test. Single-cell suspensions isolated from the lungs of animals 3 weeks post-ULDA challenge were stimulated with ID91 protein (**H**,**J**) or ID91 peptide pool (**I**,**K**) and evaluated for (**H**,**I**) CD4+ CD44+ CD154+ and (**J**,**K**) CD8+ CD44+ T cell expression of IFNγ, IL-2 and TNF by flow cytometry. *n* = 4 per group. All data are representative of two independent repeated experiments. Cytokine expression was compared using a One-Way ANOVA and Dunnet’s multiple comparisons test. * *p* < 0.05, ** *p* < 0.01 and **** *p* < 0.0001.

**Table 1 vaccines-11-00130-t001:** ID91 peptide pool composition used for peptide stimulation of splenocytes.

**Pool 1**		43	AAAYETAYRLTVPPP	86	PGLPVEYLQVPSPSM
1	HMTINYQFGDVDAHG	44	YRLTVPPPVIAENRT	87	LQVPSPSMGRDIKVQ
2	FGDVDAHGAMIRAQA	45	PVIAENRTELMTLTA	88	MGRDIKVQFQSGGNN
3	GAMIRAQAGSLEAEH	46	TELMTLTATNLLGQN	89	QFQSGGNNSPAVYLL
4	AGSLEAEHQAIISDV	47	ATNLLGQNTPAIEAN	90	NSPAVYLLDGLRAQD
5	HQAIISDVLTASDFW	48	NTPAIEANQAAYSQM	**Pool 10**	
6	VLTASDFWGGAGSAA	49	NQAAYSQMWGQDAEA	91	LDGLRAQDDYNGWDI
7	WGGAGSAACQGFITQ	50	MWGQDAEAMYGYAAT	92	DDYNGWDINTPAFEW
8	ACQGFITQLGRNFQV	**Pool 6**		93	INTPAFEWYYQSGLS
9	QLGRNFQVIYEQANA	51	AMYGYAATAATATEA	94	WYYQSGLSIVMPVGG
10	VIYEQANAHGQKVQA	52	TAATATEALLPFEDA	95	SIVMPVGGQSSFYSD
**Pool 2**		53	ALLPFEDAPLITNPG	96	GQSSFYSDWYSPACG
11	AHGQKVQAAGNNMAQ	54	APLITNPGGLLEQAV	97	DWYSPACGKAGCQTY
12	AAGNNMAQTDSAVGS	55	GGLLEQAVAVEEAID	98	GKAGCQTYKWETFLT
13	QTDSAVGSSWA DDID	56	VAVEEAIDTAAANQL	99	YKWETFLTSELPQWL
14	SSWA DDIDWDAIAQC	57	DTAAANQLMNNVPQA	100	TSELPQWLSANRAVK
15	DWDAIAQCESGGNWA	58	LMNNVPQALQQLAQP	**Pool 11**	
16	CESGGNWAANTGNGL	59	ALQQLAQPAQGVVPS	101	LSANRAVKPTGSAAI
17	AANTGNGLYGGLQIS	60	PAQGVVPSSKLGGLW	102	KPTGSAAIGLSMAGS
18	LYGGLQISQATWDSN	**Pool 7**		103	IGLSMAGSSAMILAA
19	SQATWDSNGGVGSPA	61	SSKLGGLWTAVSPHL	104	SSAMILAAYHPQQFI
20	NGGVGSPAAASPQQQ	62	WTAVSPHLSPLSNVS	105	AYHPQQFIYAGSLSA
**Pool 3**		63	LSPLSNVSSIANNHM	106	IYAGSLSALLDPSQG
21	AAASPQQQIEVADNI	64	SSIANNHMSMMGTGV	107	ALLDPSQGMGPSLIG
22	QIEVADNIMKTQGPG	65	MSMMGTGVSMTNTLH	108	GMGPSLIGLAMGDAG
23	IMKTQGPGAWPKCSS	66	VSMTNTLHSMLKGLA	109	GLAMGDAGGYKAADM
24	GAWPKCSSCSQGDAP	67	HSMLKGLAPAAAQAVE	110	GGYKAADMWGPSSDP
25	SCSQGDAPLGSLTHI	68	PAAAQAVETAAENGV	**Pool 12**	
26	PLGSLTHILTFLAAE	69	ETAAENGVWAMSSLG	111	MWGPSSDPAWERNDP
27	ILTFLAAETGGCSGS	70	VWAMSSLGSQLGSSL	112	PAWERNDPTQQIPKL
28	ETGGCSGSRDD VVDF	**Pool 8**		113	PTQQIPKLVANNTRL
29	SRDD VVDFGALPPEI	71	GSQLGSSLGSSGLGA	114	LVANNTRLWVYCGNG
30	FGALPPEINSARMYA	72	LGSSGLGAGVAANLG	115	LWVYCGNGTPNELGG
**Pool 4**		73	AGVAANLGRAASVGS	116	GTPNELGGANIPAEF
31	INSARMYAGPGSASL	74	GRAASVGSLSVPPAW	117	GANIPAEFLENFVRS
32	AGPGSASLVAAAKMW	75	SLSVPPAWAAANQAV	118	FLENFVRSSNLKFQD
33	LVAAAKMWDSVASDL	76	WAAANQAVTPAARAL	119	SSNLKFQDAYNAAGG
34	WDSVASDLFSAASAF	77	VTPAARALPLTSLTS	120	DAYNAAGGHNAVFNF
35	LFSAASAFQSVVWGL	78	LPLTSLTSAAQTAPG	**Pool 13**	
36	FQSVVWGLTVGSWIG	79	SAAQTAPGHMLGGLP	121	GHNAVFNFPPNGTHS
37	LTVGSWIGSSAGLMA	80	GHMLGGLPLGHSVNA	122	FPPNGTHSWEYWGAQ
38	GSSAGLMAAAASPYV	**Pool 9**		123	SWEYWGAQLNAMKGD
39	AAAASPYVAWMSVTA	81	PLGHSVNAGSGINNA	124	QLNAMKGDLQSSLGA
40	VAWMSVTAGQAQLTA	82	AGSGINNALRVPARA	125	AMKGDLQSSLGAGKL
**Pool 5**		83	ALRVPARAYAIPRTP		
41	AGQAQLTAAQVRVAA	84	AYAIPRTPAAG FSRP		
42	AAQVRVAAAAYETAY	85	PAAG FSRPGLPVEYL		

Peptides composing ID91 fusion antigen: Rv3619 (brown), Rv2389 (green), Rv3478 (blue), and Rv1886 (red).

**Table 2 vaccines-11-00130-t002:** Immunogenicity and Efficacy using Prime-Boost regimens.

Candidate Regimen		CFU Log10 ± SEM		CFU Log10 Reduction ^$^		Log10 ID91 Total IgG EPT ^$^	% ID91 CD4+ T cells ^#^	% ID91 CD8+ T cells ^#^
Prime	Boost	Lung	Spleen	Lung	Spleen			
Saline	Saline	5.36 ± 0.09	4.08 ± 0.48	-	-	2.18	0.398	0.304
ID91 + GLA-SE	ID91 + GLA-SE	4.61 ± 0.15	3.52 ± 0.63	0.743 ** *p* = 0.0075	0.557 *p* = 0.9638	8.25 **** *p* < 0.0001	1.209 *p* = 0.3415	1.198 *p* = 0.8055
ID91-RNA + NLC	ID91-RNA + NLC	5.05 ± 0.017	4.51 ± 0.74	0.306 *p* = 0.4731	−0.427 *p* = 0.9883	4.98 **** *p* < 0.0001	0.212 *p* = 0.9944	1.512 *p* = 0.9952
ID91 + GLA-SE	ID91-RNA + NLC	4.93 ± 0.11	4.58 ± 0.39	0.428 *p* = 0.1920	−0.501 *p* = 0.9765	6.09 **** *p* < 0.0001	0.748 *p* = 0.9216	0.994 *p* = 0.9573
ID91-RNA + NLC	ID91 + GLA-SE	4.51 ± 0.18	3.72 ± 0.57	0.847 ** *p* = 0.0021	0.358 *p* = 0.9947	6.87 **** *p* < 0.0001	1.088 *p* = 0.4890	1.110 *p* = 0.9945
Combination	Combination	4.54 ± 0.13	2.66 ± 0.76	0.809 ** *p* = 0.0047	1.419 *p* = 0.4273	7.45 **** *p* < 0.0001	1.711 * *p* = 0.0487	3.048 * *p* = 0.0226

^$^ Vaccine cohort mean values compared to the saline group; significance evaluated by using One-way ANOVA with Dunnett’s multiple-comparison test denoted by asterisk. CFU standard error for lung = 0.219, spleen = 0.938. ^#^ Magnitude of Polyfunctional T-cell responses include IFNγ, TNF and IL-2 cytokines in % ID91-specific T-cell responses. Data representative of two independent repeat studies for CFU, and EPT, data representative of one repeat for T cell data. *n* = 7–6/group for CFU, *n* = 4/group for EPT and T cell evaluations. * *p* < 0.05, ** *p* < 0.01 and **** *p* < 0.0001.

## Data Availability

Data are reported in the manuscript or can be made available upon request from the authors.

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
