# Peer review of "An RNA-Based Vaccine Platform for Use against Mycobacterium tuberculosis"

_vaccines, 2023, doi:10.3390/vaccines11010130_

Round 1

Reviewer 1 Report

With the recent success of mRNA vaccines against SARS-CoV-2 and the need for more efficient and affordable tuberculosis vaccines, the replicating-RNA (repRNA) system holds great promise for future vaccine production. In their manuscript, Larsen et al. are studying the immunogenicity and the prophylactic efficacy of a repRNA-based ID91 vaccine against Mycobacterium tuberculosis (M.tb). To this end, the authors have generated a self-replicating RNA construct for the intracellular expression of ID91 (a previously described tuberculosis vaccine candidate), and they show that an in vitro transfection of their construct into BHK cells produces a protein that can be detected using mouse sera obtained from ID91-protein immunized animals. The RNA construct is then evaluated by its ability to restrict low-dose M.tb infection and to elucidate humoral (Ig-levels) and cellular (T cell proliferation, cytokine secretion of T cells) immune responses in vivo in mice in comparison to the protein-based ID91 vaccine (+ GLA-SE adjuvant) and saline. While some prophylactic efficacy as well as both elevated IgG-titers and ID91-specific IL2-secreting T cells are observed with repRNA vaccine in comparison to the saline treated controls, the traditional protein-based vaccine seems to elicit better protection and a stronger immune response. Finally, the assessment of the prophylactic efficacy against an ultra-low dose M.tb infection demonstrates some prophylactic protection with the 2x ID91 protein, with the heterologous RNA>protein vaccination and in the so called “combination” vaccine group but not in the 2x ID91 RNA immunized mice. While the manuscript is well-written and justified and the introduced repRNA-vaccine platform timely and interesting, there are some concerns that would be important to assess prior publication.

Major comments:

- The current title concludes that the RNA-based vaccine has prophylactic efficacy against M.tb. While this conclusion is supported by the one immunization/low dose aerosol (LDA) result shown in Figure 2D, this seems not to be supported by the data of the 2xRNA-immunization/ultra-low dose aerosol (ULDA) experiment (Table 1 and Figure 3A). In fact, since the authors consider the ULDA challenge physiologically more relevant, the title doesn´t fully reflect the obtained results and should be reconsidered.

In their abstract, the authors mention the usage of both the fusion protein vaccine and the repRNA vaccine. While this is correct, for this reviewer the abstract gives the impression that the ID91+GLA-SE mixture is also a novel vaccine candidate to be tested. However, since the authors have already previously studied the effects of the protein-based ID91+GLA-SE in their tuberculosis model (

- Figure 1A shows a schematic structure of ID91 with an expected size of 90 kDa, whereas Figure 1C demonstrates a Western-Blot analysis with a bright band of approximately 60kDa and a dim band of 30kDa in the cell lysate of ID91 transfected BHK cells. Please, explain the size discrepancy between the expected and detected sizes. Preferably, a positive control ID91 protein should be ran and displayed on the same gel/blot to rule out electrophoresis artefacts and to better demonstrate that the repRNA-encoding protein is of correct size and that the mouse serum used for detection identifies truly ID91. To further improve the scientific transparency of the data in panel 1C, both the SDS-PAGE gel image and the blot could be shown side-by-side.

- For evaluation of the flow cytometry data, gating strategies and some representative flow cytometry plots (e.g. for the repRNA group) should be demonstrated for Figures 1D, 2A-B and 3G-H either in the figures or as supplementary information. This is important since multiple parameters such as proper compensation, cell counts within gates and the gates themselves can have major implications to the obtained data.

- Although the prophylactic effects of the ID91-encoding repRNA vaccine seems modest, the manuscript provides compelling evidence on the immunogenicity of this vaccine. In fact, the immunogenic properties could be better highlighted in the discussion. In addition, the reasons behind the prophylactic effects of the heterologous RNA>Protein vaccine and not the Protein>RNA group could be speculated in more detail in the discussion.

Minor comments:

- There are many acronyms in the manuscript such as LMIC (line 41), DR (line 43), MDR (45), COP (line 60) that are not used later in the text. Please, remove unnecessary acronyms from the text to improve the readability. On the other hand, since EPT (line 174), BALf (line 175) and CFU (line 183) are used in several places these should be deciphered.

- Line 120 states that both non-reducing and reducing conditions were used for SDS-PAGE and Western-Blotting but Figure 1C shows only one condition without specifying if this is non-reducing or reducing. Please, address this matter.

- Line 179: The authors are likely referring to HRP-conjugated anti-Ig antibodies. Please, modify.

-´ Line 214: expressioin -> expression

- Lines 311 and 313: ID93 vs. ID91. Please, modify.

- In Figure 1D, it makes sense that the scales of the axes describing the frequencies of the cells are different for different cytokines. However, within a given parameter such as TNFa also the axes between CD4+ and CD8+ T cells differ. Since this makes data comparison difficult, unifying these axes scales would help in data interpretation.

- In Figures 2D and 3A the y-axis doesn´t start at 0. While this is understandable to visualize the modest overall differences in the logarithmic bacterial loads, it can easily give false impression about the data. Consequently, the y-axis could be modified to start at 0 or alternatively the figure legends could state that the axis doesn´t start at zero.

- While the timepoints used for immunizations and boosts and the M.tb challenge has been described in the Materials and Methods section, using schematics e.g. in each figure would aid in interpreting the data and in understanding the differences between the experiments. There is also discrepancy between lines 135-136 and 224; the other states that the immunization were done three weeks apart and the other that they were done two weeks apart, respectively.

- Table 1 (Combination, CFU Log10- lung burden) has the average CFU of 3.90+/-0.66. However, in the Figure 3A the average CFU is around 4.50. Please, correct.

- Please, explain why CD154 is used as an additional marker to distinguish the activated CD4+  T cells in the Figure 3G. In fact, showing cytokine positive CD4+ T cells that are only CD44+ would make comparison to the CD8+CD44+ T cells (Figure 3H) easier.

- In Figure 3G, the authors display the flow cytometry results of the cytokine production of the CD4+ T cells stimulated with the complete ID91 protein, whereas in Figure 3H results after a peptide pool stimulation are shown. For this reviewer, the rationale behind this choice is unclear. In fact, it would be interesting to see and compare the effects of both the whole protein and the peptide pool stimulation on both the CD4+ and CD8+ T cells. Also, the peptide pool used for the stimulation should be described as in Figure 1D.

- The differences between the low- and ultra-low dose challenged repRNA immunized mice could be speculated. In other words, why there is a difference in repRNA immunized animals compared to saline controls in the low-dose model but not in the ultra-low dose model that has even got a booster repRNA vaccination?

Author Response

Point-by-point Response to Reviewer Comments

Reviewer 1

With the recent success of mRNA vaccines against SARS-CoV-2 and the need for more efficient and affordable tuberculosis vaccines, the replicating-RNA (repRNA) system holds great promise for future vaccine production. In their manuscript, Larsen et al. are studying the immunogenicity and the prophylactic efficacy of a repRNA-based ID91 vaccine against Mycobacterium tuberculosis (M.tb). To this end, the authors have generated a self-replicating RNA construct for the intracellular expression of ID91 (a previously described tuberculosis vaccine candidate), and they show that an in vitro transfection of their construct into BHK cells produces a protein that can be detected using mouse sera obtained from ID91-protein immunized animals. The RNA construct is then evaluated by its ability to restrict low-dose M.tb infection and to elucidate humoral (Ig-levels) and cellular (T cell proliferation, cytokine secretion of T cells) immune responses in vivo in mice in comparison to the protein-based ID91 vaccine (+ GLA-SE adjuvant) and saline. While some prophylactic efficacy as well as both elevated IgG-titers and ID91-specific IL2-secreting T cells are observed with repRNA vaccine in comparison to the saline treated controls, the traditional protein-based vaccine seems to elicit better protection and a stronger immune response. Finally, the assessment of the prophylactic efficacy against an ultra-low dose M.tb infection demonstrates some prophylactic protection with the 2x ID91 protein, with the heterologous RNA>protein vaccination and in the so called “combination” vaccine group but not in the 2x ID91 RNA immunized mice. While the manuscript is well-written and justified and the introduced repRNA-vaccine platform timely and interesting, there are some concerns that would be important to assess prior publication.

Major comments:

  • The current title concludes that the RNA-based vaccine has prophylactic efficacy against M.tb. While this conclusion is supported by the one immunization/low dose aerosol (LDA) result shown in Figure 2D, this seems not to be supported by the data of the 2xRNA-immunization/ultra-low dose aerosol (ULDA) experiment (Table 1 and Figure 3A). In fact, since the authors consider the ULDA challenge physiologically more relevant, the title doesn´t fully reflect the obtained results and should be reconsidered.
    • Response: we agree with the reviewer and have adjusted our title to better reflect the nature of our adapting this platform for Mtb and less about the efficacy demonstrated by this specific vaccine candidate. “An RNA-based vaccine platform for use against Mycobacterium tuberculosis

  • In their abstract, the authors mention the usage of both the fusion protein vaccine and the repRNA vaccine. While this is correct, for this reviewer the abstract gives the impression that the ID91+GLA-SE mixture is also a novel vaccine candidate to be tested. However, since the authors have already previously studied the effects of the protein-based ID91+GLA-SE in their tuberculosis model (https://doi.org/10.4049/jimmunol.1401103), modifications to the abstract should be made to clarify whether the protein vaccine was used as a positive control or if new aspects of this vaccine were also studied. This comment also applies to the Results section (lines 223-224).
    • Response: We changed the first mention of ID91 in the abstract to better reflect that ID91 protein vaccine is a second generation vaccine candidate and that the RNA platform use is the novel aspect. In lines 223-224 we have also made these changes and added the original reference for ID91 antigen fusion.

  • Figure 1A shows a schematic structure of ID91 with an expected size of 90 kDa, whereas Figure 1C demonstrates a Western-Blot analysis with a bright band of approximately 60kDa and a dim band of 30kDa in the cell lysate of ID91 transfected BHK cells. Please, explain the size discrepancy between the expected and detected sizes. Preferably, a positive control ID91 protein should be ran and displayed on the same gel/blot to rule out electrophoresis artefacts and to better demonstrate that the repRNA-encoding protein is of correct size and that the mouse serum used for detection identifies truly ID91. To further improve the scientific transparency of the data in panel 1C, both the SDS-PAGE gel image and the blot could be shown side-by-side.
    • Response: Given our current western uses polyclonal sera we are confident that the band being detected is expression of ID91 fusion antigen and follow-on studies in the manuscript also suggest this fusion antigen is being produced by the repRNA (namely the epitope mapping experiment where we see responses across the antigen in overlapping peptide pools). However, since protein control is most compelling we repeated the blot and included ID91 protein and a more appropriate ladder range as well. We are now happy to show in Figure 1 and Supplementary Figure 1 that the product of ID91 RNA in BHK cells matches the most intense band from the ID91 protein lane. Thank you for this suggestion, we agree this improves the manuscript.

  • For evaluation of the flow cytometry data, gating strategies and some representative flow cytometry plots (e.g. for the repRNA group) should be demonstrated for Figures 1D, 2A-B and 3G-H either in the figures or as supplementary information. This is important since multiple parameters such as proper compensation, cell counts within gates and the gates themselves can have major implications to the obtained data.
    • Response: We agree and have added supplemental figures 2, 3, and 4 including representative gating schemes for each of the experimental data sets shared. We apologize for the oversight of this omission.

  • Although the prophylactic effects of the ID91-encoding repRNA vaccine seems modest, the manuscript provides compelling evidence on the immunogenicity of this vaccine. In fact, the immunogenic properties could be better highlighted in the discussion. In addition, the reasons behind the prophylactic effects of the heterologous RNA>Protein vaccine and not the Protein>RNA group could be speculated in more detail in the discussion.
    • Response: Thank you for this positive review of the platform and for the suggestion to more thoroughly highlight the immunogenicity in the discussion, this has been added.

Minor comments:

  • There are many acronyms in the manuscript such as LMIC (line 41), DR (line 43), MDR (45), COP (line 60) that are not used later in the text. Please, remove unnecessary acronyms from the text to improve the readability. On the other hand, since EPT (line 174), BALf (line 175) and CFU (line 183) are used in several places these should be deciphered.
    • Response: Thank you for this note, these acronyms have been expanded or detailed as suggested. LMIC is on line 41 and 92 and has remained abbreviated, but others have been corrected.

  • Line 120 states that both non-reducing and reducing conditions were used for SDS-PAGE and Western-Blotting but Figure 1C shows only one condition without specifying if this is non-reducing or reducing. Please, address this matter.
    • Response: This was an error in the methods, the conditions are only non-reducing and we have made this correction.

  • Line 179: The authors are likely referring to HRP-conjugated anti-Ig antibodies. Please, modify.
    • Response: Thank you, this has been corrected.

  • -´ Line 214: expressioin -> expression
    • Response: Thank you, this has been corrected.

  • Lines 311 and 313: ID93 vs. ID91. Please, modify.
    • Response: Thank you, these have been corrected to ID91.

  • In Figure 1D, it makes sense that the scales of the axes describing the frequencies of the cells are different for different cytokines. However, within a given parameter such as TNFa also the axes between CD4+ and CD8+ T cells differ. Since this makes data comparison difficult, unifying these axes scales would help in data interpretation.
    • Response: Thank you for this suggestion, we have modified the heat-map to be more uniform in its display and we agree this makes comparisons much simpler.

  • In Figures 2D and 3A the y-axis doesn´t start at 0. While this is understandable to visualize the modest overall differences in the logarithmic bacterial loads, it can easily give false impression about the data. Consequently, the y-axis could be modified to start at 0 or alternatively the figure legends could state that the axis doesn´t start at zero.
    • Response: This clarification has been added to the corresponding figure legends.

  • While the timepoints used for immunizations and boosts and the M.tb challenge has been described in the Materials and Methods section, using schematics e.g. in each figure would aid in interpreting the data and in understanding the differences between the experiments. There is also discrepancy between lines 135-136 and 224; the other states that the immunization were done three weeks apart and the other that they were done two weeks apart, respectively.
    • Response: Thank you for this suggestion. We have added schematics to each of the three figures with data, depicting the different regimens used and timepoints being evaluated. We have also corrected the description error noted in timing.

  • Table 1 (Combination, CFU Log10- lung burden) has the average CFU of 3.90+/-0.66. However, in the Figure 3A the average CFU is around 4.50. Please, correct.
    • Response: Thank you for bringing our attention to this error. The error was in the Table CFU value, the correct value is 4.54 +/- 0.13 and the reduction values have also been verified and are correct as written. It was a transcription mistake and as such all values were subsequently cross checked in the table and are confirmed to be accurate.

  • Please, explain why CD154 is used as an additional marker to distinguish the activated CD4+ T cells in the Figure 3G. In fact, showing cytokine positive CD4+ T cells that are only CD44+ would make comparison to the CD8+CD44+ T cells (Figure 3H) easier.
    • Response: Thank you for this question. There are several TB-field specific references which highlight the importance of CD154 as a marker of antigen-specific CD4+ T cell responses but no such corresponding evidence to our knowledge for its use in CD8+ T cells. Based on these references described below and others we use CD154 to try and capture the most-likely high quality antigen-experienced CD4+ T cells induced by vaccination and/or infection.
      • PMID: 29065175 “Comparative analysis of activation induced marker (AIM) assays for sensitive identification of antigen-specific CD4 T cells” where CD154 is used as a clear activation marker for CD4+ T cells in this assay.
      • PMID: 35618277 “Mycobacterium tuberculosis-specific CD4 T-cell scoring discriminates tuberculosis infection from disease” where CD154 is used in human phenotyping of CD4+ activated cells in the context of human infection and disease states.
      • PMID: 28821584 “Transcriptome analysis of mycobacteria-specific CD4+ T cells identified by activation induced expression of CD154”, which shows these cells are enriched for cytokine and activation responses.

  • In Figure 3G, the authors display the flow cytometry results of the cytokine production of the CD4+ T cells stimulated with the complete ID91 protein, whereas in Figure 3H results after a peptide pool stimulation are shown. For this reviewer, the rationale behind this choice is unclear. In fact, it would be interesting to see and compare the effects of both the whole protein and the peptide pool stimulation on both the CD4+ and CD8+ T cells. Also, the peptide pool used for the stimulation should be described as in Figure 1D.
    • Response: We have now added a table describing the peptide pools in the materials and methods and we agree this adds clarity. We have also added a better description of pools associated with specific responses in the epitope mapping experiment. We have also included a two additional panels to Figure 4 which include peptide pool stimulation for CD4+ T cells as well as protein stimulation for CD8+ T cells so that these data are available for comparisons. Thank you for this suggestion.

  • The differences between the low- and ultra-low dose challenged repRNA immunized mice could be speculated. In other words, why there is a difference in repRNA immunized animals compared to saline controls in the low-dose model but not in the ultra-low dose model that has even got a booster repRNA vaccination?
    • Response: We believe that this is likely due to the interval of immunizations (e.g. close together) rather than that nature of the challenge dose. This speculation is based on some work on the repRNA platform that has not yet been published as well as our 2020 STM manuscript and clinical work with the protein-adjuvant vaccine candidate ID93. We have included some of these hypotheses in our discussion/conclusions, but ensured they are clearly speculative at this point and require follow up.

Reviewer 2 Report

Larsen et al. have provided a pre-clinical proof of concept study of an RNA-based vaccine for Mycobacterium tuberculosis in order to map some CD4+ and CD8+ T cell responses. Notably, the authors compare T cell response profiles of RNA and protein-based vaccines. Indeed, each vaccine has elicited unique CD4+ and CD8+ T cell responses. Further, a single immunization from either vaccine has revealed moderate prophylactic protection against M. tuberculosis H37Rv compared to unvaccinated controls (~0.5log10 in lung titers). Prime-boost vaccination strategies, however, have shown enhanced prophylactic protection against infection compared to those unvaccinated animals (~1log10 in lung titers). This has been evident in animal groups receiving either a prime-boost of the protein vaccine, an RNA vaccine followed by a protein-boost vaccine, or the combination of both vaccines. While antibody responses have largely followed a trend among different prime-boost vaccination strategies, the CD8+ CD44+ T cell response in the combination vaccination has been significantly dominated by TNF expression. Overall, this study has shown that combination RNA and protein vaccination strategies could enhance prophylactic treatment against M. tuberculosis.

General comments:

The authors have revealed preliminary profiles of T cell responses to RNA and protein-based vaccines. As noted in the discussion, much work is required to optimize the formulation and timing of prime-boost vaccines in order to further assess the B and T cell responses. Of note, developing a robust mucosal immunity through intranasal vaccines has recently been proposed for some respiratory diseases. The authors would ideally further discuss this aspect with regards to whether one of their vaccines could be administered intranasally.

Further, it would be relevant to include a table listing the composition of each peptide pool. Additionally, while the materials and methods section briefly describes the overlap between a peptide and some 8 amino acids of the ID91 fusion peptide, it would be more insightful to include a schematic mapping the amino acids covered by each peptide or peptide pool.  

Author Response

Point-by-point Response to Reviewer Comments

Reviewer 2:

Larsen et al. have provided a pre-clinical proof of concept study of an RNA-based vaccine for Mycobacterium tuberculosis in order to map some CD4+ and CD8+ T cell responses. Notably, the authors compare T cell response profiles of RNA and protein-based vaccines. Indeed, each vaccine has elicited unique CD4+ and CD8+ T cell responses. Further, a single immunization from either vaccine has revealed moderate prophylactic protection against M. tuberculosis H37Rv compared to unvaccinated controls (~0.5log10 in lung titers). Prime-boost vaccination strategies, however, have shown enhanced prophylactic protection against infection compared to those unvaccinated animals (~1log10 in lung titers). This has been evident in animal groups receiving either a prime-boost of the protein vaccine, an RNA vaccine followed by a protein-boost vaccine, or the combination of both vaccines. While antibody responses have largely followed a trend among different prime-boost vaccination strategies, the CD8+ CD44+ T cell response in the combination vaccination has been significantly dominated by TNF expression. Overall, this study has shown that combination RNA and protein vaccination strategies could enhance prophylactic treatment against M. tuberculosis.

General comments:

  • The authors have revealed preliminary profiles of T cell responses to RNA and protein-based vaccines. As noted in the discussion, much work is required to optimize the formulation and timing of prime-boost vaccines in order to further assess the B and T cell responses. Of note, developing a robust mucosal immunity through intranasal vaccines has recently been proposed for some respiratory diseases. The authors would ideally further discuss this aspect with regards to whether one of their vaccines could be administered intranasally.
    • Response: Thank you for this suggestion, we have added comments about the intranasal administration of our protein-GLA-SE vaccines to-date and have stated the work for i.n. or aerosol delivery of RNA platform is ongoing.

  • Further, it would be relevant to include a table listing the composition of each peptide pool. Additionally, while the materials and methods section briefly describes the overlap between a peptide and some 8 amino acids of the ID91 fusion peptide, it would be more insightful to include a schematic mapping the amino acids covered by each peptide or peptide pool.

Response: We agree this is an important aspect to include. In the materials and methods, we have added a table (Table 1) listing all of the peptides generated across the ID91 fusion as well as identified which peptides have come from which of the 4 Mtb antigens in the fusion. In addition, we now reference specific peptide responses in the discussion of peptide stimulation results.

Round 2

Reviewer 1 Report

The authors have properly addressed all the previously raised concerns.

Please, double check correct spelling of the revised sections. Some examples of existing typos are shown below:

-          Row 240: exsisting

-          Row 242: prophilactic

-          Row 258: restimuation

-          Row 267: Preotein

-          Row 433: ability "to" stimulate

-          Row 468: enagge

Author Response

We would like to again, thank the reviewer for their suggestions as we feel the manuscript has greatly improved. 

All of these typos are addressed and corrected in the revised manuscript accompanying this reply. WE have also performed a final spelling and grammar check of the full document.
